# Elevated Blood Alcohol Concentration Is Associated with Improved Clinical Outcomes of Intravenous Thrombolysis Treatment in Acute Ischemic Stroke Patients—A Retrospective Study

**DOI:** 10.3390/jcm12062238

**Published:** 2023-03-14

**Authors:** Tamás Árokszállási, Eszter Balogh, Rita Orbán-Kálmándi, Máté Pásztor, Anita Árokszállási, Edit Boglárka Nagy, Ivett Belán, Zsolt May, Tünde Csépány, László Csiba, Zsuzsa Bagoly, László Oláh

**Affiliations:** 1Department of Neurology, Faculty of Medicine, Doctoral School of Neuroscience, University of Debrecen, 22 Móricz Zsigmond krt, H-4032 Debrecen, Hungary; 2Division of Clinical Laboratory Sciences, Department of Laboratory Medicine, Faculty of Medicine, University of Debrecen, 98 Nagyerdei krt, H-4032 Debrecen, Hungary; 3Department of Neurology, Medical Centre, Hungarian Defence Forces, 44 Károly Róbert krt, H-1134 Budapest, Hungary; 4Department of Oncology, Faculty of Medicine, University of Debrecen, 98 Nagyerdei krt, H-4032 Debrecen, Hungary; 5Division of Radiology and Imaging Science, Department of Medical Imaging, Faculty of Medicine, University of Debrecen, 98 Nagyerdei krt, H-4032 Debrecen, Hungary; 6ELKH-DE Cerebrovascular Research Group, 22 Móricz Zsigmond krt, H-4032 Debrecen, Hungary

**Keywords:** stroke, observational study, thrombolysis, ethanol, outcomes

## Abstract

Background: Intravenous thrombolysis (IVT) improves acute ischemic stroke (AIS) outcomes, but with limited success. In addition, ethanol potentiates the effect of r-tPA in ischemia models. Methods: The effect of acute alcohol consumption on IVT outcomes was investigated in a retrospective cohort study. AIS patients with detectable blood alcohol concentration (BAC) during IVT were included (alcohol group; *n* = 60). For each case, 3 control subjects who underwent IVT but denied alcohol consumption were matched in terms of age, sex, affected brain area, and stroke severity. Outcomes were determined using the NIHSS at 7 days and the modified Rankin scale (mRS) at 90 days. Results: Patients were younger and had a less severe stroke than in a standard stroke study. Favorable long-term outcomes (mRS 0–2) occurred significantly more frequently in the alcohol group compared to controls (90% vs. 63%, *p* < 0.001). However, the rates of hemorrhagic transformation were similar. Multiple logistic regression models identified elevated BAC as a significant protective factor against unfavorable short-term (OR: 0.091, 95% CI: 0.036–0.227, *p* < 0.001) and long-term outcomes (OR: 0.187, 95% CI: 0.066–0.535, *p* = 0.002). In patients with BAC > 0.2%, significantly lower NIHSS was observed at 3 and 7 days after IVT vs. in those with 0.01–0.2% ethanol levels. Conclusion: Elevated BAC is associated with improved outcomes in IVT-treated AIS without affecting safety.

## 1. Introduction

Stroke is among the leading causes of death and disability worldwide [1]. In acute ischemic stroke (AIS) without large vessel occlusion, intravenous thrombolysis (IVT) remains the mainstay pharmacological therapy. Although IVT significantly increased the rate of excellent outcomes at 3 months in the NINDS and ECASS-3 studies, its therapeutic success remains limited for the majority of patients [2,3]. The combination of IVT and neuroprotective treatment has been the focus of attention as a potential therapy to improve the effectiveness of IVT [4,5]. Lemarchand et al. found that 6 weeks of alcohol consumption in a murine model influenced the hepatic clearance of tPA, leading to worsening of the ischemic lesion and a lack of beneficial effects of tPA induced thrombolysis [6]. However, acute administration of ethanol in a transient focal cerebral ischemia model was reported to provide a strong neuroprotective effect and did not increase the risk of intracerebral hemorrhage when used in combination with thrombolytic agents [4,5]. Although the precise mechanism responsible for the favorable effect of alcohol is still unknown, effects related to beneficial hemodynamic and hemostasis alterations as well as antiapoptotic effects have been reported [7,8,9,10,11]. Moreover, ethanol may inhibit NMDA-stimulated excitotoxicity, which likely contributes to tissue damage in ischemic stroke and head injury [12].

A pilot study in humans also demonstrated the neuroprotective effects of small doses of ethanol. In a retrospective cohort study, patients with severe traumatic brain injury and low blood alcohol concentration (BAC) at admission were reported to have lower mortality as compared to patients with no or high BAC [13]. Two pilot trials demonstrated that intravenous administration of caffeinol (ethanol and caffeine) alone or combined with IVT is safe and feasible in acute stroke patients. However, these studies included very few patients (≤25); moreover, no control group was used, therefore the effect of caffeinol on IVT outcome could not be evaluated [14,15].

Based on the intriguing experimental results listed above, our aim was to investigate the effect of acute alcohol consumption on the outcome of IVT in AIS patients in a retrospective observational cohort study. 

## 2. Materials and Methods

A retrospective observational cohort study of consecutively admitted AIS patients treated with IVT was conducted. The study was based on a large, prospectively collected thrombolysis registry and the electronic medical records of two large Hungarian stroke centers (the Department of Neurology, University of Debrecen, Debrecen, and the Department of Neurology, Military Hospital—State Health Centre, Budapest). The study protocol was approved by the Ethics Committee of the Hungarian Medical Research Council (Registration number: 50362-2/2019/EKU). Detailed electronic medical records of IVT-treated AIS patients admitted between 1 January 2010 and 1 September 2019 were screened to identify cases with alcohol consumption just before the stroke event and a detectable BAC at admission. A BAC was obtained in all cases of visible or suspected alcohol intake. For each case, 3 control subjects who underwent IVT due to AIS but denied consuming alcohol for 24 h before admission were chosen from the databases. The controls were matched in terms of age, sex, affected brain area, and severity of their neurological symptoms at admission. The inclusion and exclusion criteria of cases and controls were identical to the standard criteria for recombinant tissue plasminogen activator (r-tPA) administration according to the current European Stroke Organization (ESO) guidelines at the time of the event [16]. Additional exclusion criteria included diseases influencing long-term outcomes (malignancy, liver failure, and renal failure). All cases and controls underwent IVT within the 4.5 h therapeutic time window using r-tPA (Boehringer Ingelheim, Germany) according to standard protocols [16]. Baseline clinical data retrieved included demographics, comorbidities, medications, histories of cerebrovascular and cardiovascular diseases, BMI, cerebrovascular risk factors, details of IVT treatment (onset-to-needle time (OTN), door-to-needle time (DTN), r-tPA dose), detailed laboratory reports (clinical chemistry tests including electrolytes, glucose, liver- and renal function tests, C-reactive protein measurement, complete blood count, hemostasis panel, BAC) from admission blood samples, and imaging results. At admission, all patients underwent cerebral CT and extra- and intracranial CT-angiography. A follow-up cerebral CT was performed for every patient 24 h after treatment. CT images taken on admission and 24 h after IVT were analyzed by two independent radiologists, and Alberta Stroke Program Early CT Scores (ASPECTS) were calculated [17]. Hemorrhagic transformation was classified according to the European Cooperative Acute Stroke Study (ECASS) II criteria on follow-up CT scans performed 24 h after IVT [18].

As part of the prospective registry, a questionnaire was used to assess the cerebrovascular risk factors of patients, including alcohol consumption habits. In this questionnaire, patients (or patients assisted by relatives) were asked about the frequency, type, and amount of their weekly alcohol consumption. Alcohol intake was calculated and expressed in grams of ethanol per week. Based on the amount of alcohol consumed, patients were classified as non-drinkers (0 g/week), mild (less than 105 g/week), moderate (at least 105 but less than 210 g/week), and heavy drinkers (at least 210 g/week) [19]. Elevated serum gamma-glutamyl transferase (GGT) levels (>50 U/L) together with increased erythrocyte mean cellular volume (MCV) values (>100 fL) were considered indirect signs of chronic alcohol consumption [20]. 

AIS subtypes were classified according to the modified TOAST (Trial of ORG 10,172 in Acute Stroke Treatment) criteria [21]. Stroke severity was determined by the National Institutes of Health Stroke Scale (NIHSS) on admission, at 2, 24, 72 h after IVT, and day 7. All cases and controls were followed, and the modified Rankin Scale (mRS) score was recorded 3 months after the stroke event [22]. 

The following outcomes and safety endpoints were investigated in the study: The short-term outcome was assessed by the NIHSS score 7 days after IVT. Favorable outcome was defined as a decrease of at least 4 points in the NIHSS score as compared to the pretreatment value or a decrease to 0 points, while poor outcome was defined as an increase of at least 4 points [2].The long-term outcome was assessed by the mRS score at 3 months after stroke onset. Patients with an mRS score of 0–2 were defined as having a favorable long-term outcome [23].The safety of IVT in both groups was evaluated by the occurrence of intracranial hemorrhage according to the ECASS II criteria, based on the presence of asymptomatic (aSICH) or symptomatic intracranial hemorrhage (SICH).Mortality was assessed at 3 months post-event.

Statistical analysis was performed using the statistical package for social sciences (SPSS, Release 26.0, Chicago, IL, USA) and GraphPad Prism 8.0 (GraphPad Prism Inc., La Jolla, CA, USA). The Shapiro–Wilk test was used to assess the normality of the data. A student’s t test or Mann–Whitney U test was performed for independent two-group analyses. ANOVA with Bonferroni post hoc test or Kruskal–Wallis analysis with Dunn–Bonferroni post hoc test was applied for multiple comparisons. Differences between categorical variables were assessed by χ^2^-test or Fisher’s exact test where appropriate. Binary backward logistic regression models were used to determine the independent effect of acute alcohol consumption on short- and long-term IVT outcomes. The adjustments of the models were based on the results of preliminary statistical analyses of baseline characteristics between groups (Student’s *t*-test or Mann-Whitney U-test, χ^2^-test, or Fisher’s exact), literature data, and methodological principles. Results of the logistic regression analysis were expressed as odds ratios (OR) and 95% confidence intervals (CI). Shift analysis on the mRS score at 3 months after IVT was performed using an ordinal regression model. A *p*-value of <0.05 was considered statistically significant.

## 3. Results

In the investigated period of 104 months, 2876 AIS patients received IVT treatment in the two stroke centers. Of these patients, 144 were reported to have consumed alcohol within the past 12 h preceding their admission, of whom 64 had a detectable BAC at admission and were included in the study (cases). Two cases were excluded a posteriori due to pulmonary and gastrointestinal malignancies diagnosed after IVT. As most patients who consumed alcohol were either ineligible for mechanical thrombectomy or the procedure was unavailable at the time, only two cases were found that were treated with mechanical thrombectomy after IVT. To ensure the homogeneity of the groups, these cases were not included in the final analysis. Baseline characteristics and outcomes of cases (alcohol group, *n* = 60) and controls (*n* = 180) are shown in Table 1. 

Hypertension was less common, and the body mass index (BMI) was significantly lower in the alcohol group as compared to the control group. The prevalence of other major cerebrovascular risk factors did not differ between the two groups. Due to the lower BMI in the alcohol group, the r-tPA dose administered during IVT was significantly lower in the alcohol group as compared to the control group. Medication at admission was similar in the two groups except for antiplatelet treatment, which was less frequently used in the alcohol group. Stroke subtypes based on TOAST criteria, onset-to-needle time (OTN), and door-to-needle time (DTN) did not differ between groups. As expected, in the alcohol group, serum glucose concentrations were significantly lower, while serum aspartate aminotransferase (AST), GGT activities, and erythrocyte MCV were significantly higher as compared to controls. The ratio of moderate and heavy drinkers was significantly higher in the alcohol group as compared to controls (73% vs. 40%, *p* < 0.001). 

As acute alcohol consumption may cause neurological signs, symptoms related to acute alcohol consumption, including dysarthria, ataxia, and eye-movement disorder, were analyzed separately within the first 24 h in both groups (Table 2). 

As expected, dysarthria was significantly more common in the alcohol group as compared to the controls; however, the occurrence rates of eye-movement disorder and ataxia did not significantly differ between groups. When evaluating potentially alcohol-related symptoms at 24 h after IVT as compared to baseline, dysarthria was the only symptom showing significant improvement in patients who drank alcohol before admission as compared to controls. It must be noted, however, that improvement of dysarthria, as measured by a maximum 2-point decrease in the NIHSS, occurred in only 7 cases in the alcohol group (data not shown in Table); therefore, the diminishing alcohol-related neurological symptoms are presumed to have a relatively low contributory effect on the outcomes investigated in this study. 

Although baseline NIHSS scores were identical in the two investigated groups, significantly lower NIHSS scores were found in the alcohol group at 24 h, 72 h, and 7 days after IVT (Table 1). The extent of neurological improvement calculated as the decrease in NIHSS value was significantly higher in the alcohol group as compared to controls at all investigated time points. The extent of the decrease was most prominent after the 24 h post-IVT time points and reached a median of −6 points in NIHSS value in the alcohol group (median: −6 IQR [−7 to −4] vs. −2 [−5 to 0] points in controls; *p* < 0.001), indicating that the significantly better neurological improvement observed in the alcohol group cannot be attributed to the diminishing effect of alcohol. 

In the control group, the percentage of patients with a favorable short-term outcome was 36%, which is in line with data derived from large clinical trials (Table 1). Remarkably, in the alcohol group, the percentage of good outcomes was 82%, a highly significant difference from the controls (36%; *p* < 0.001). Favorable long-term outcome, defined as a 90-day mRS score of 0–2, was also significantly more common in the alcohol group compared with the controls (90% versus 63%, *p* < 0.001). A significant shift towards good functional outcomes was found in the alcohol group as compared to controls, with a prominent difference in patients having mRS 0 (45% vs. 19%, respectively; Figure 1).

The difference between groups was not significant in terms of the frequency of intracranial hemorrhage after IVT (Table 1). The mortality rate at 90 days after the event did not differ between groups (Table 1). 

To find out which parameters are associated with the short- and long-term outcomes of IVT in the investigated cohort, a binary backward logistic regression analysis was used (Table 3). Results of univariate analysis revealing parameters that significantly influence short- and long-term outcomes are shown in Appendix A. According to binary backward logistic regression models including all relevant parameters (based on the results of univariate analysis, adjusting for history of chronic alcohol consumption and laboratory results suggestive of chronic alcohol consumption), acute alcohol consumption was identified as a significant protective factor against unfavorable short- (OR: 0.091, 95% CI: 0.036–0.227, *p* < 0.001) and long-term outcomes (OR: 0.187, 95% CI: 0.066–0.535, *p* = 0.002) (Table 3). 

The statistical model presented for short-term outcomes included the adjustment for dysarthria at admission and dysarthria at 24 h in order to omit effects attributed to alcohol-related neurological improvement; however, practically identical results were obtained when these parameters were not included in the applied model (data not shown). 

To find out whether BAC influences the clinical outcome of IVT, patients in the alcohol group were divided into two subgroups: patients with detectable admission BAC below (n = 44) and above (n = 16) 0.2%. Baseline NIHSS scores did not differ between the two subgroups at admission or at 2 h and 24 h after IVT (Figure 2A–C, respectively). 

On the other hand, at 72 h, significantly lower NIHSS scores were observed in patients with BAC > 0.2% as compared to those with lower detectable alcohol levels (median [IQR]: 3 (0–7) vs. 1 (0–3), respectively, *p* = 0.045, Figure 2D) and 7 days after IVT (median [IQR]: 2 (0–7) vs. 1 (0–2), respectively, *p* = 0.049, Figure 2E). Admission BAC did not show a significant association with long-term outcomes as assessed by mRS at 90 days at this sample size (Figure 2F, *p* = 0.095). It must be noted, however, that all patients with a BAC of >0.2% had good long-term outcomes (mRS 0–2), with a median value of mRS 0. 

## 4. Discussion

While the association between regular alcohol intake and the risk of stroke has been studied in a number of reports, the effect of acute alcohol intake on the outcome of stroke has not yet been thoroughly addressed in clinical investigations. In this study, for the first time, we show evidence that acute alcohol consumption results in a significantly higher rate of favorable short- and long-term outcomes in AIS patients treated with IVT (82% versus 36% in controls, and 90% versus 63% in controls, respectively, *p* < 0.001), without a significant difference in intracranial hemorrhage or mortality rate. In binary backward logistic regression models, acute alcohol consumption before the onset of stroke was found to confer a strong independent protective effect against poor short- and long-term IVT outcomes, revealing an effect independent of chronic alcohol intake. Moreover, within the alcohol group, a dose-response effect of ethanol concentration was found on short-term outcomes, as a BAC of >0.2% was found to provide a more pronounced beneficial effect compared to a lower BAC. 

Alcohol may produce neurological signs, which may improve as alcohol is eliminated and may be incorrectly attributed to the favorable effect of IVT. Therefore, alcohol-related symptoms were analyzed separately in this study. Of all potential alcohol-related symptoms, dysarthria was more common and showed significant improvement within the first 24 h in the alcohol group as compared to the controls. Although this was a confounding factor in evaluating the effect of acute alcohol consumption on outcomes, statistically, the much larger difference in NIHSS scores observed at 24 h and 72 h after IVT between the alcohol and control groups cannot be attributed to this single symptom. Results of the backward logistic regression analysis revealed no major difference when the symptom of dysarthria at admission and at 24 h was included or excluded from the model; therefore, the diminishing alcohol-related neurological symptoms are presumed to have a relatively low contributory effect on the outcomes investigated in this study. 

In the literature, controversy exists on the effect of chronic alcohol intake and the risk of stroke, while little is known about the effect of regular drinking on stroke outcomes [24]. Moderate alcohol intake was not associated with the risk of AIS in most studies, while a few studies and a meta-analysis concluded that light to moderate drinking (≤2 drinks/day) is associated with a reduced risk of AIS, while heavy drinking (>4 drinks/day) is associated with an increased risk of both AIS and hemorrhagic stroke [24]. Others also reported that chronic heavy alcohol drinking has an aggravating effect on the outcome of ischemic stroke [25,26]. It must be noted, however, that discriminating between the causal effects of alcohol and lifestyle-related risks of stroke has been a challenge for these studies. As of today, the relationship between chronic alcohol intake and functional outcomes of stroke is sparse; few studies, including a large prospective cohort study, have shown no association between drinking habits and functional outcomes of stroke, although patients receiving IVT have not been studied separately [27,28]. Here we confirm the findings of earlier reports that drinking habits are less likely to be contributing factors to AIS outcomes. In this study, as expected, the ratio of moderate and heavy drinkers (at least 105 g alcohol/week) was significantly higher in the alcohol group as compared to controls, and indirect signs of chronic alcohol consumption were significantly higher in the alcohol group as well. As an attempt to minimize bias related to self-reported information, in addition to the reported alcohol consumption habits, indirect laboratory signs of chronic alcohol consumption (GGT and MCV of erythrocytes) were also considered in the statistical models based on previous literature [20]. Although in the univariate model, GGT and alcohol consumption habits showed significant associations with short-term outcomes of IVT, no such association was seen for long-term outcomes. As alcohol consumption habits or indirect laboratory signs of chronic alcohol consumption did not remain in the backward multiple regression analysis as significant parameters associated with outcomes, we believe that the effect of chronic alcoholism on improved outcomes is negligible in this cohort, and the data support a more prominent role for acute alcohol consumption.

The reason for the beneficial effect of acute alcohol consumption on IVT outcomes in AIS patients is an intriguing question. In this study, a dose-response beneficial effect of ethanol concentration was found on short-term outcomes, which implies direct, ethanol-related pathophysiology. The time course of the alcohol-related beneficial effect is also intriguing. Based on the comparison of NIHSS at different time points after thrombolysis between cases and controls, neurological improvement attributed to the effect of alcohol is mainly observed after 24 h post-lysis and is most prominent within the first week after the event. At 24 h post-lysis, fewer radiologic signs of early ischemic changes were found in the alcohol group as compared to controls, as slightly but significantly higher ASPECTS were found at this time point in the alcohol group (median [IQR]: 9 (7–10) vs. 8 (6–9) in controls). In theory, ethanol may improve AIS outcomes by decreasing the extent of neurological damage and/or by potentiating the effect of thrombolysis. One possible explanation for the first may be that low-moderate dose of alcohol produces vasodilation in the cerebral resistance vessels [7]. Vasodilation might improve the cerebral blood flow in the penumbra region, which can decrease the size of the irreversibly damaged brain tissue. As for the second theory, alcohol might beneficially influence the balance of hemostasis in the case of an acute thrombotic event: by inhibiting platelet aggregation and increasing fibrinolytic activity, it may potentiate the efficacy of thrombolysis [8,9,10]. It is also known that ethanol may inhibit NMDA-induced excitotoxicity in neuronal culture, which may play an important role in contributing to the pathomechanism of acute ischemic stroke [12]. In addition, the antiapoptotic effect of ethanol has been reported in experimental focal cerebral ischemia models after intravenous administration of ethanol, which could also contribute to the favorable outcomes observed in AIS [4,5,11]. Further research is warranted on the pathophysiology behind the observed beneficial effects of acute alcohol intake on AIS IVT outcomes.

Although some studies reported that ethanol may produce microhemorrhages and large intracerebral hemorrhages in rats [29,30], hemorrhagic side effects, including the hemorrhagic transformation of ischemic lesions, did not occur more frequently in the alcohol group compared to the control group in our study. These results are in line with the results of previous human pilot trials, providing evidence that intravenous administration of ethanol is safe in AIS patients treated with IVT [14,15].

Similar to all retrospective clinical investigations, the results of this study should be interpreted in the context of its limitations. First, although patients or patients’ relatives denied acute alcohol intake in the control group, the possibility of recent alcohol consumption could not be fully ruled out for each subject in this group, as BAC was not measured routinely in these centers unless there were visible signs or a suspicion of alcohol intake. Second, despite the screening period of 9.75 years in two large stroke centers, the number of patients in the alcohol group was relatively low. Third, the amount and type of alcohol consumed varied, resulting in different BACs, potentially leading to a heterogenous effect. Fourth, as discussed in detail above, alcohol-related neurological symptoms may be a confounding factor in this study design. Fifth, patients were younger in this study as compared to the average age of stroke patients [31], which can be explained by the fact that heavy drinking increases 1.7–2.5 times the risk of ischemic stroke between 50 and 65 years old and many shorten the time to stroke by about 5 years [32]. Accordingly, previous studies have reported a significantly lower age in heavy drinking-stroke patients [25,26,33]. Sixth, patients had a less severe stroke in this study than in most randomized, controlled trials of AIS treated with IVT. While in the majority of these trials, patients had a baseline NIHSS score of 12–14 points and an average age of 67–69 years, the baseline NIHSS score was only 8 points and the median age was only 62 years in our control group [2,34]. These differences might explain why the rate of the favorable outcome defined by mRS 0–2 was relatively high (63%) in our control group and even higher in the alcohol group (90%). 

## 5. Conclusions

In conclusion, in this retrospective observational cohort study, elevated BAC was identified as a significant protective factor against unfavorable short- and long-term outcomes of stroke in IVT-treated AIS patients, without influencing safety. A dose-response effect of BAC on short-term outcomes was also observed in the investigated cohort. Based on the findings, a randomized, controlled trial is warranted to investigate the effect of ethanol on the thrombolysis outcome of AIS under standardized conditions.

## Figures and Tables

**Figure 1 jcm-12-02238-f001:**
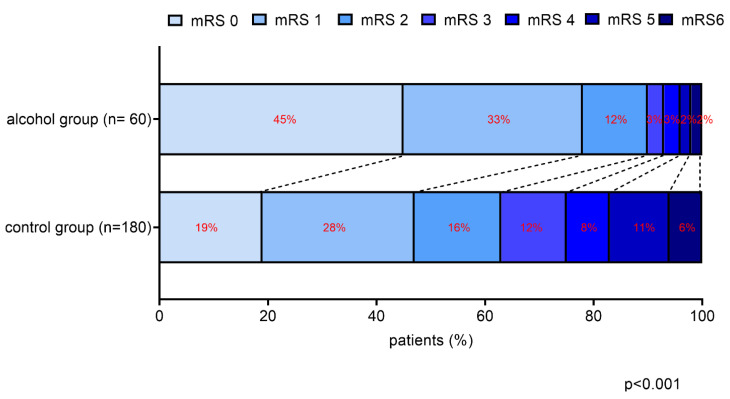
Functional outcomes at 90 days after intravenous thrombolysis in cases (alcohol group) and controls. The modified Rankin scale (mRS) at 90 days post-event demonstrates a significant shift towards good functional outcomes (mRS 0–2) in the alcohol group vs. controls (*p* < 0.001).

**Figure 2 jcm-12-02238-f002:**
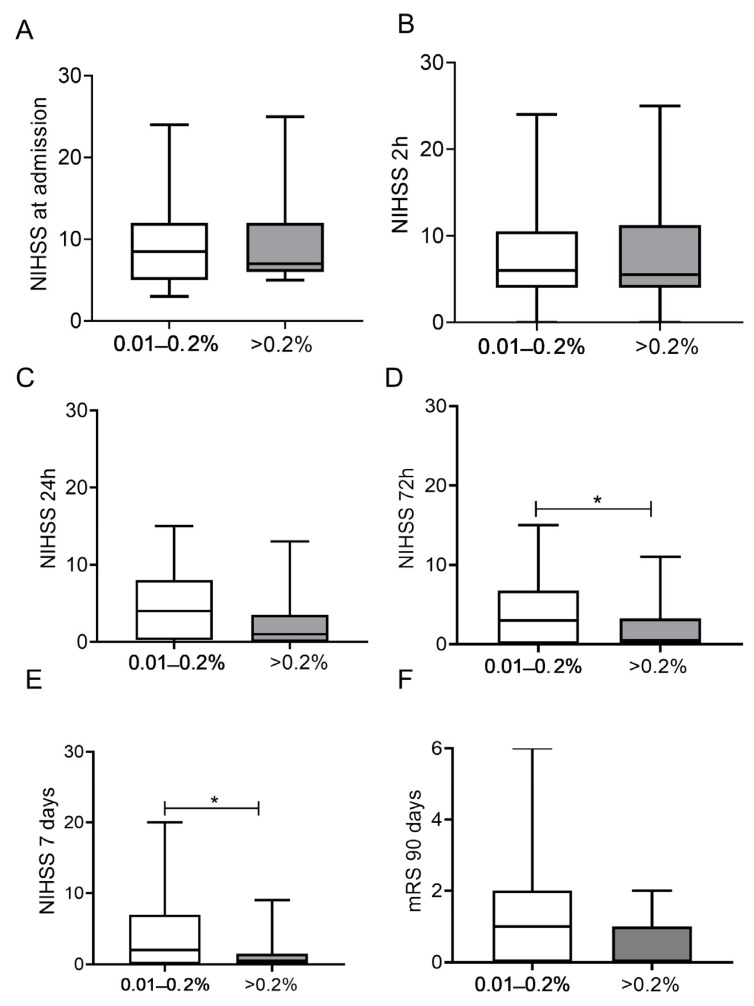
Stroke severity (NIHSS) on admission (**A**), at 2 h (**B**), at 24 h (**C**), at 72 h (**D**), at 7 days (**E**) and mRS at 90 days (**F**) in the alcohol subgroups of below and above 0.2% baseline blood alcohol concentration. NIHSS: National Institutes of Health Stroke Scale, mRS: modified Rankin Scale * *p* < 0.05.

**Table 1 jcm-12-02238-t001:** Baseline characteristics and outcome of cases (alcohol group) and controls.

	Alcohol Group	Control Group	*p*
Number of patients, *n* (%)	60	180	
Male, *n* (%)	51 (85)	155 (86)	0.975
Age (years), median (IQR)	61 (54–69)	62 (54–69)	0.893
Ethanol level (%), median (IQR)	0.1515 (0.0523–0.2045)	-	
Affected brain area, *n* (%)			
VBI	16 (26)	48 (27)	
Dominant MCA area	14 (24)	43 (24)	0.996
Non-dominant MCA area	30 (50)	89 (49)	
Cerebrovascular risk factors, *n* (%)			
Hypertension	34 (57)	145 (81)	<0.001
Diabetes mellitus	7 (12)	38 (21)	0.356
Hyperlipidemia	27 (45)	76 (39)	0.756
Atrial fibrillation	5 (8)	22 (12)	0.408
History of stroke	10 (17)	21 (11)	0.317
History of myocardial infarction	7 (12)	18 (10)	0.366
Peripheral arterial disease	2 (3)	3 (2)	0.601
Current smoker	35 (58)	85 (47)	0.136
BMI, median (IQR)	25.2 (22.1–29.3)	28.3 (24.2–31.7)	0.001
Medication at admission, *n* (%)			
Antiplatelet therapy	7 (12)	51 (28)	0.008
Oral anticoagulant	2 (3)	10 (6)	0.735
Oral antidiabetics	5 (8)	29 (16)	0.150
Lipid lowering therapy	8 (13)	28 (15)	0.751
OTN, min, median (IQR)	139.5 (113.0–177.0)	137.0 (109.0–180.0)	0.700
DTN, min, median (IQR)	41.0 (32.0–57.0)	42.5 (32.0–57.0)	0.893
r-tPA dose, mg, median (IQR)	66.0 (56.0–78.0)	76.0 (62.1–86.0)	0.002
Laboratory parameters at admission, median (IQR)			
Serum sodium (mmol/L)	140.0 (137.0–142.0)	139.0 (137.0–141.0)	0.223
Serum glucose (mmol/L)	5.8 (5.0–6.3)	6.4 (5.6–7.8)	<0.001
Creatinine (μ mol/L)	72.0 (63.8–83.0)	76.0 (65.0–91.0)	0.117
hsCRP (mg/L)	2.6 (1.1–5.8)	2.5 (1.4–4.9)	0.775
AST (U/L)	23.5 (17.0–35.0)	20.0 (15.0–24.0)	0.003
ALT (U/L)	18.5 (13.0–30.0)	19.0 (14.0–26.0)	0.809
GGT (U/L)	51.0 (28.0–101.5)	37.0 (22.5–63.0)	0.014
WBC (G/L)	7.6 (6.4–9.0)	8.4 (6.7–10.5)	0.057
MCV (fL)	92.69 ± 6.62	88.93 ± 5.69	<0.001
Platelet count (G/L)	237 (191–278)	216 (183–252)	0.055
APTT	28.9 ± 3.3	28.2 ± 3.2	0.157
INR	0.96 (0.92–1.01)	0.97 (0.93–1.01)	0.633
Stroke etiology (TOAST), n (%)			
Large artery atherosclerosis	20 (35)	58 (33)	
Cardioembolic	9 (15)	34 (20)	0.867
Small-vessel occlusion	14 (25)	44 (25)	
Other/undetermined	15 (25)	38 (22)	
Admission BP systolic (mmHg), mean±SD	160 ± 27.0	170.0 ± 26.5	0.011
Admission BP diastolic (mmHg), mean ± SD	94.3 ± 16.6	94.0 ± 17.0	0.874
ASPECTS at admission	10 (9–10)	10 (9–10)	0.649
ASPECTS at 24 h	9 (7–10)	8 (6–9)	0.017
Alcohol consumption habits, n (%)			
Non-drinker, mild drinker	16 (27)	108 (60)	<0.001
Moderate, heavy drinker	44 (73)	72 (40)	
NIHSS at admission, median (IQR)	7 (5–12)	8 (5–11)	0.699
NIHSS at 2 h, median (IQR)	6 (4–11)	7 (4–11)	0.740
NIHSS at 24 h, median (IQR)	4 (0–7)	6 (3–12)	<0.001
NIHSS at 72 h, median (IQR)	3 (0–6)	5 (3–10)	0.001
NIHSS at 7 day, median (IQR)	2 (0–7)	5 (2–10)	<0.001
ΔNIHSS 2 h–0 h	−1 (−3–0)	0 (−2–0)	0.026
ΔNIHSS 24 h–0 h	−4 (−7 to −2)	−1 (−3 to 1)	<0.001
ΔNIHSS 72 h–0 h	−5 (−7 to −3)	−2 (−5 to 0)	<0.001
ΔNIHSS day 7–0 h	−6 (−7 to −4)	−2 (−5 to 0)	<0.001
Outcomes, n (%)			
Short-term outcome (7 days)			
good outcome (NIHSS ≥ 4 points decrease or 0)	49 (82)	65 (36)	
unchanged status (ΔNIHSS ± 3 points)	8 (13)	97 (54)	<0.001
poor outcome (NIHSS ≥ 4 points increase)	3 (5)	18 (10)	
Long-term outcome (90 days)			
mRS 0–2	54 (90)	114 (63)	<0.001
mRS 3–6	6 (10)	66 (37)	
Intracranial hemorrhage (ECASS II)			
no hemorrhage	59 (98)	171 (95)	
aSICH	1 (2)	7 (4)	0.500
SICH	0 (0)	2 (1)	
Mortality at 90 days after thrombolysis	1 (2)	10 (6)	0.300

Unless otherwise indicated, data are medians (interquartile ranges) or numbers (percentage). IQR, interquartile range; VBI, vertebrobasilar insufficiency; MCA, middle cerebral artery; BMI, body mass index; ONT, onset-to-needle time; DTN, door-to-needle time; r-tPA, recombinant tissue plasminogen activator; hsCRP, high-sensitive C-reactive protein; AST, aspartate aminotransferase; ALT, alanine aminotransferase; GGT, gamma-glutamyl transferase; WBC, white blood cell count; MCV, mean corpuscular volume of red blood cells; APTT, activated partial thromboplastin time; INR, international normalized ratio; TOAST, Trial of ORG 10,172 in Acute Stroke Treatment; BP, blood pressure; SD, standard deviation; ASPECTS, The Alberta stroke program early CT score; NIHSS, National Institutes of Health Stroke Scale; mRS, modified Rankin Scale; ECASS II, European Co-operative Acute Stroke Study-II; aSICH, asymptomatic intracranial hemorrhage; SICH, symptomatic intracranial hemorrhage.

**Table 2 jcm-12-02238-t002:** Frequency of neurological signs related to acute alcohol consumption (dysarthria, ataxia, and eye movement disorder) before and after thrombolysis in cases (alcohol group) and controls.

	Alcohol Group (n = 60)	Control Group (n = 180)	*p*
Patients with presence of symptoms, *n* (%)			
At admission			
dysarthria	42 (70)	96 (53)	0.024
ataxia	10 (17)	23 (13)	0.449
eye movement disorder	15 (25)	39 (22)	0.592
At 24 h post-event			
dysarthria	22 (37)	80 (44)	0.291
ataxia	6 (10)	20 (11)	0.811
eye movement disorder	6 (10)	31 (17)	0.179
Ratio of patients with improvement of symptoms, *n*/total, (%)			
At 24 h post-event			
dysarthria	25/42 (60)	20/96 (21)	<0.001
ataxia	7/10 (70)	12/23 (52)	0.341
eye movement disorder	11/15 (73)	18/39 (46)	0.073

Data are numbers (percentage). Improvement of symptoms were defined as a decrease in the respective NIHSS category score from 2 to 1, from 2 to 0 or from 1 to 0.

**Table 3 jcm-12-02238-t003:** Independent predictors for unfavorable short- and long-term outcomes of intravenous thrombolysis in the investigated cohort.

	OR	95% CI	*p*
Unfavorable short-term outcome ^£^			
Acute alcohol consumption	0.091	0.036–0.227	<0.001
NIHSS on admission (per 1 point)	0.908	0.844–0.976	0.009
Age (per 1 year)	1.035	1.004–1.067	0.025
Systolic blood pressure on admission (per 1 Hgmm)	1.015	1.002–1.028	0.025
AST (per 1 U/L)	0.978	0.955–1.003	0.082
Unfavorable long-term outcome ^§^			
NIHSS on admission (per 1 point)	1.215	1.121–1.318	<0.001
Acute alcohol consumption	0.187	0.066–0.535	0.002
Age (per 1 year)	1.056	1.015–1.097	0.006
Systolic blood pressure on admission (per 1 Hgmm)	1.017	1.004–1.031	0.013
Serum glucose on admission (per 1 mmol/L)	1.125	0.999–1.267	0.051
White blood cell count on admission (per 1 G/L)	1.139	0.986–1.315	0.077

Last step of backward multiple regression analysis is provided. ^£^ Unfavorable short-term outcome is defined as a less than 4 points decrease of NIHSS and the NIHSS > 0 by day 7 post-event. Backward multiple regression models included age, sex, acute alcohol consumption, hypertension, antidiabetic therapy, lipid lowering therapy, NIHSS at admission, CRP at admission, AST at admission, GGT at admission, MCV at admission (threshold > 100 fL), admission systolic and diastolic blood pressure, alcohol consumption habits (non- or mild drinker vs. moderate or heavy drinker), presence of dysarthia at admission and 24 h. ^§^ Unfavorable long-term outcome is defined as mRS 3–6 by day 90 post-event. Backward multiple regression models included age, sex, acute alcohol consumption, hypertension, diabetes mellitus, NIHSS on admission, admission serum glucose level, CRP at admission, WBC at admission, AST at admission, GGT at admission, MCV at admission (threshold > 100 fL), admission systolic and diastolic blood pressure, alcohol consumption habits (non- or mild drinker vs. moderate or heavy drinker).

## Data Availability

The data presented in this study are available on request from the corresponding author.

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
