# Peer review of "Elevated Blood Alcohol Concentration Is Associated with Improved Clinical Outcomes of Intravenous Thrombolysis Treatment in Acute Ischemic Stroke Patients—A Retrospective Study"

_jcm, 2023, doi:10.3390/jcm12062238_

Round 1
Reviewer 1 Report
Title: Acute alcohol consumption improves the outcome of intravenous thrombolysis treatment in acute ischemic stroke patients.
This article presents assessment of the concept. Acute alcohol consumption improves the outcome of intravenous thrombolysis treatment in acute ischemic stroke patients. This article takes an appropriates starting approach and resulted in relative clinical finding.
The literature review is little but covering the main relevant theories and empirical studies in this area. It reflects a sound grasp of the literature and a well-argued critical perspective on this material. I would suggest covering the relevant area and references. The candidate presents a clear account of her conceptual approach to critique and demonstrates a reflexive perspective on work. The article provides a good balance between a critique of assumptions and a detailed empirical investigation.
Article presents a well-structured and clear account of the methodological and results framework of the research, the rationale for the research design and a description of the methods used. Coverage of the reasons for shifting the focus of the research question through the course of the research is appropriate and clearly argued. In general, the coverage of the development of the methodology, the research design and results are thoughtful and coherent.
The key contributions of the research well, presenting some thoughtful reflections on the research process and future directions for work in this area. I found it interesting to see the work developed in this way, and in my view this article makes a very important contribution to our understanding.
The article is well-written, with very few clerical errors, and the style and layout are good. I had a few suggested minor revisions in term of sentence framing and typos. The candidate clearly demonstrates creative abilities in his research field and the article meets the required standard
Author Response
Reply to Reviewer#1
Dear Reviewer,
We greatly appreciate your efforts and time to evaluate our manuscript (jcm-2217924), entitled „Acute alcohol consumption improves the outcome of the intravenous thrombolysis treatment in acute ischemic stroke patients”. Each comment has been carefully considered and the manuscript was revised accordingly. Point-by-point responses are indicated in bold and italics below. The Introduction and the References were improved as suggested. The English grammar and style of the manuscript were revised by a certified paid English editing service.
Comment:
The literature review is little but covering the main relevant theories and empirical studies in this area. It reflects a sound grasp of the literature and a well-argued critical perspective on this material. I would suggest covering the relevant area and references.
According to your and Reviewer#2’s suggestions, additional references were added to the Introduction and Discussion sections of the manuscript (references 11, 28-32).
Thank you again for your time to review our work. We believe that the changes introduced in the manuscript resulted in an improved version of the paper. We hope that the revised manuscript will be found acceptable for publication in the ’Journal of Clinical Medicine’.
Yours sincerely,
Prof. László Oláh
corresponding author
olah@med.unideb.hu

Reviewer 2 Report
The article by Arokszallasi et al. aimed at investigating the effect of acute alcohol consumption on intravenous thrombolysis (IVT) outcomes from data retrospectively obtained in a cohort of acute ischemic stroke (AIS) patients. Cases (n=60) were selected from AIS patients with detectable blood alcohol concentration (BAC) during IVT. Controls (n=180) were AIS patients who underwent IVT and denied consuming alcohol for 24 hours before admission. The authors found favourable long-term outcomes (mRS 0-2) more frequently in the alcohol group compared with controls. Multiple logistic regression models identified acute alcohol consumption as significant protective factor against unfavourable short-term and long-term outcomes.
The article is well written and of interest in the field. However, in the opinion of this reviewer, there are some important drawbacks in the design of the study:
- - Controls were selected based on their declaration (no BAC measured). BAC from control patients is necessary to accurately compare the effects of acute alcohol on stroke outcome (patients with positive alcohol levels vs patients with undetectable levels of alcohol as well as different BAC dose-effects).
- - As the authors mention in the discussion, average age of stroke patients in this study (61 years old) is abnormally low compared to standard age for stroke (71 years old in men; Akyea et al., Stroke 2021). This could be explained by the fact that heavy drinking increases 1.7-2.5x the risk of ischemic stroke between 50 and 65 years old and may shorten time to stroke by ≈5 years regardless of familial and other common confounds (Kadlekova et al., Stroke 2015). Accordingly, previous studies have reported significantly lower age in heavy drinking-stroke patients (59 years old vs 72 in controls, Ducroquet et al., Stroke, 2013; 66 vs 75 years old in Gattringer et al., Neurology 2015; and 67 vs 73 years old in Drieu et al., JCI Insight, 2020).
Given that controls were chosen to match the alcohol group, it is then normal that the average age of the control group was also low. This means that the present study does not represent average stroke patients. Consequently, the authors should highlight this fact a particularity of the study design, and needs to be mentioned from the beginning (even in the title) as a “retrospective study in young stroke patients” or something similar.
- - In the opinion of this reviewer, “acute alcohol consumption” in the title is misleading, because it does not take into account chronic, past alcohol consumption. I would suggest the authors to change “acute alcohol consumption” by “positive blood alcohol levels” or something similar.
- Alcohol withdrawal is not taken into account as a potential confounding factor in moderate/heavy drinkers (who represent 40% and 73% of patients in the control and alcohol group respectively). Indeed, 40% of patients in the control group were in alcohol withdrawal at the moment of their stroke (based on their declaration), whereas moderate/heavy drinkers patients in the alcohol group were not at withdrawal (BAC+).
- - Past, chronic alcohol consumption is also not taken into account in this study. However, previous data have reported worse stroke outcomes in heavy drinkers (Ducroquet et al., Stroke, 2013; Drieu et al., JCI Insight, 2020). It would be necessary to compare the subgroups of patients depending on their alcohol consumption habits (non-drinker, mild, moderate or heavy) and the BAC (positive or negative).
- - I would suggest to include in the introduction/discussion the anti-glutamatergic effects of ethanol (that could block AIS-induced excitotoxicity) as a neuroprotective mechanism that could explain the effects found in the alcohol group.
Author Response
Reply to Reviewer#2
Dear Reviewer,
We greatly appreciate your efforts and time to evaluate our manuscript (jcm-2217924), entitled „Acute alcohol consumption improves the outcome of the intravenous thrombolysis treatment in acute ischemic stroke patients”. Each comment has been carefully considered and the manuscript was revised accordingly. Point-by-point responses are indicated in bold and italics below. The Introduction and the References were improved as suggested. The English grammar and style of the manuscript were revised by a certified paid English editing service.
Comments:
1/ Controls were selected based on their declaration (no BAC measured). BAC from control patients is necessary to accurately compare the effects of acute alcohol on stroke outcome (patients with positive alcohol levels vs patients with undetectable levels of alcohol as well as different BAC dose-effects).
Thank you for your comment. Examination of blood alcohol content is not a routine procedure at our clinic. In case of suspicion of alcohol consumption, or if the patient reported alcohol intake, blood was taken to measure blood alcohol levels. However, if alcohol consumption was not suspected, no visible signs of alcohol intake was observed, or the patient/or relative denied alcohol intake, blood alcohol level was not measured. Due to the retrospective design of the study, blood alcohol data could not be obtained from control subjects. This issue was highlighted in the Discussion chapter as one of the limitations of our work. As we concluded, a prospective study is highly warranted in the future, which could overcome this limitation.
2/ As the authors mention in the discussion, average age of stroke patients in this study (61 years old) is abnormally low compared to standard age for stroke (71 years old in men; Akyea et al., Stroke 2021). This could be explained by the fact that heavy drinking increases 1.7-2.5x the risk of ischemic stroke between 50 and 65 years old and may shorten time to stroke by ≈5 years regardless of familial and other common confounds (Kadlekova et al., Stroke 2015). Accordingly, previous studies have reported significantly lower age in heavy drinking-stroke patients (59 years old vs 72 in controls, Ducroquet et al., Stroke, 2013; 66 vs 75 years old in Gattringer et al., Neurology 2015; and 67 vs 73 years old in Drieu et al., JCI Insight, 2020)
Thank you for your valuable comment, this is indeed a very important point raised. The explanation you suggested together with the respective references had been added to the Discussion section (lines 361-365).
3/ Given that controls were chosen to match the alcohol group, it is then normal that the average age of the control group was also low. This means that the present study does not represent average stroke patients. Consequently, the authors should highlight this fact a particularity of the study design, and needs to be mentioned from the beginning (even in the title) as a “retrospective study in young stroke patients” or something similar.
Thank you for your comment. The title was supplemented with the phrase „retrospective study”, as suggested. If possible, however, we would like to refrain from using the term „young stroke patients” in the title of the paper, as no age limit was pre-determined in our study. The younger age observed in our patients as compared to the standard age of stroke resulted from the earlier onset of stroke in the alcohol group. Therefore, the patient population and matching controls were not selected to be „young” by applying an age limit. If the young age was mentioned in the title, it may be misunderstood by the general reader as a reference to a selection of patients by their younger age, however, there was no such a selection.
4/ In the opinion of this reviewer, “acute alcohol consumption” in the title is misleading, because it does not take into account chronic, past alcohol consumption. I would suggest the authors to change “acute alcohol consumption” by “positive blood alcohol levels” or something similar.
Respecting your comment, we would like to keep the original title, because the favourable effect of acute alcohol consumption on the clinical outcome was clearly demonstrated in our study, and this effect was independent of the chronic alcohol intake.
Binary backward logistic regression models in our study included all relevant parameters potentially influencing stroke outcomes, including the history of chronic alcohol consumption and laboratory results suggestive of chronic alcohol consumption. Using these models, acute alcohol consumption was identified as a significant protective factor against unfavorable short- (OR:0.101, 95%CI:0.040-0.255, p<0.0001) and long-term outcomes (OR:0.182, 95%CI:0.062-0.535, p=0.002), (lines 223-232). As we highlight in the Discussion section of the manuscript (lines 269-281 and 313-320), the beneficial effect of acute alcohol consumption on outcomes was independent of chronic alcohol intake, as alcohol consumption habits or indirect laboratory signs of chronic alcohol consumption did not remain in the backward multiple regression analysis as significant parameters associated with outcomes. We believe that the effect of chronic alcoholism on improved outcomes is negligible in this cohort and data support a more prominent role of acute alcohol consumption. Thus, we believe that acute alcohol consumption is directly related to the positive outcomes observed in this cohort, while other terms, including „positive blood alcohol levels” may generate confusion in the general reader as this terminology is somewhat obscure and it is not recommended according to the Clinical and Laboratory Standards Institute (CLSI) for blood alcohol testing in the clinical laboratory.
5/ Alcohol withdrawal is not taken into account as a potential confounding factor in moderate/heavy drinkers (who represent 40% and 73% of patients in the control and alcohol group respectively). Indeed, 40% of patients in the control group were in alcohol withdrawal at the moment of their stroke (based on their declaration), whereas moderate/heavy drinkers patients in the alcohol group were not at withdrawal (BAC+).
The medical history of the patients included in this study was carefully checked again, and predelirium state occurred in less than 10% of patients in both groups without developing delirium (predelirium developed in 4 patients /6.7%/ in the alcohol and 10 patients /5.6%/ in the control group). Patients already in the predelirium stage (recognized by increased perspiration, increased pulse rate, and tremor) were treated with chlordiazepoxide, beta blockers, and tiapride as needed. We believe that alcohol withdrawal may have affected both groups equally, as none of the patients had access to alcohol after admission.
6/ Past, chronic alcohol consumption is also not taken into account in this study. However, previous data have reported worse stroke outcomes in heavy drinkers (Ducroquet et al., Stroke, 2013; Drieu et al., JCI Insight, 2020). It would be necessary to compare the subgroups of patients depending on their alcohol consumption habits (non-drinker, mild, moderate or heavy) and the BAC (positive or negative).
This is indeed a key question in our study, and statistical analyses were made accordingly. To our surprise, data did not support the hypothesis that heavy drinkers showed worse stroke outcome. Results of univariate analysis on outcomes related to alcohol consumption habits, BAC and other baseline anthropometric data are presented in Supplementary Tables 1 and 2. According to these analyses, significantly more patients who were moderate or heavy drinkers had improved short-term outcomes, while long-term outcomes were not affected by chronic alcohol intake. In agreement with these findings, in the univariate model, indirect laboratory signs of alcohol consumption habits (increased GGT, AST) showed significant association with improved short-term outcomes of IVT, but no such association was seen for long-term outcomes. As alcohol consumption habits or indirect laboratory signs of chronic alcohol consumption did not remain in the backward multiple regression analysis as significant parameters associated with outcomes, we believe that the effect of chronic alcoholism on improved outcomes is negligible in this cohort and data support a more prominent role of acute alcohol consumption. As we understand that arguments related to these findings are crucial in our study, a separate chapter was dedicated to this topic (lines 295-320). In the literature, controversy exists on the effect of chronic alcohol intake and the risk of stroke, while little is known about the effect of regular drinking and stroke outcomes [22]. As of today, the relationship between chronic alcohol intake and functional outcomes of stroke are sparse, few studies, including a large prospective cohort study have shown no association between drinking habits and functional outcome of stroke, although patients receiving IVT have not been studied separately [23,24]. Here we confer the findings of earlier reports that drinking habits are less likely to be contributing factors to AIS outcome. In the Conclusion section of the manuscript, however, we also emphasize that future, prospective studies are warranted to investigate the effect of acute and chronic ethanol intake on acute ischemic stroke IVT outcomes (lines 372-378).
7/ I would suggest to include in the introduction/discussion the anti-glutamatergic effects of ethanol (that could block AIS-induced excitotoxicity) as a neuroprotective mechanism that could explain the effects found in the alcohol group.
Thank you very much for your comment. This is an excellent point raised and it is very much appreciated. The anti-glutamatergic effects of ethanol are now included in the Introduction and the Discussion sections of the paper (lines 54-56 and 338-341).
Thank you again for your thorough review of our work and the important comments raised. We believe that the changes introduced in the manuscript resulted in an improved version of the paper. We hope that the revised manuscript will be found acceptable for publication in the ’Journal of Clinical Medicine’.
Yours sincerely,
László Oláh
corresponding author
olah@med.unideb.hu

Reviewer 3 Report
Following corrections should be incorporated.
Line 81” What is “rt-PA”. It should be described in full (recombinant tissue plasminogen activator) and put in bracket for the first time it comes in manuscript. Later it can be used.
rt-PA should be written as r-tPA.
In the same way, describe ESO in full and put in bracket if it appears later, otherwise a full description followed by in bracket (ESO) is needed.
Line 147: Convert “uneligible” into “ineligible.”
Line 169: Change the word “Admission” into “Administered.”
Line 331: Correct “vessels6.”
Author Response
Reply to Reviewer#3
Dear Reviewer,
We greatly appreciate your efforts and time to evaluate our manuscript (jcm-2217924), entitled „Acute alcohol consumption improves the outcome of the intravenous thrombolysis treatment in acute ischemic stroke patients”. Each comment has been carefully considered and the manuscript was revised accordingly. Point-by-point responses are indicated in bold and italics below. The Introduction and the References were improved as suggested. The English grammar and style of the manuscript were revised by a certified paid English editing service.
Comments:
Page 2
Materials and methods
- Line 81: ” What is “rt-PA”. It should be described in full (recombinant tissue plasminogen activator) and put in bracket for the first time it comes in manuscript. Later it can be used. rt-PA should be written as r-tPA. In the same way, describe ESO in full and put in bracket if it appears later, otherwise a full description followed by in bracket (ESO) is needed.
As requested, rt-PA was described in full and later used as r-tPA throughout the manuscript. Similarly, all abbreviations including ESO was described in full at first appearance.
Page 4
Results
- Line 147: Convert “uneligible” into “ineligible.”
Uneligible was converted to inegligible as requested.
Page 6
Results
Line 169: Change the word “Admission” into “Administered.”
The ambiguous term ’Admission medication’ was corrected to ’Medication at admission’.
Page 12
Discussion
Line 331: Correct “vessels6.”
The typo was corrected.
Thank you again for your review of our work. We believe that the changes introduced in the manuscript resulted in an improved version of the paper. We hope that the revised manuscript will be found acceptable for publication in the ’Journal of Clinical Medicine’.
Yours sincerely,
Prof. László Oláh
corresponding author
olah@med.unideb.hu

Round 2
Reviewer 2 Report
- - As already mentioned in my first reviewing, I consider “acute alcohol consumption” in the title as misleading. “Positive blood alcohol content” seems to me a more accurate term. This is extremely relevant from a public health point of view.
- - Still in the title, the term “improves” is, at least, an over interpretation and should be modified by “is associated with”.
- - For me, the studied population is very different in terms of age and stroke severity from the most randomized, controlled trials on AIS and this fact needs to be mentioned, at least in the title and/or the abstract.
- - There are mistakes in the OR values given in the abstract and the results sections concerning factors protecting against unfavourable short (OR:0.101) and long-term outcomes (0.182) compared to the values given in table 3 (0.091 and 0.187 respectively). Please look over.
- - Alcohol consumption habits were significantly associated with the short-term outcome (supplementary table 1, p. 14) (by the way, it is not clear whether the significance (p=0.003) concerns “non-, mild drinkers” or “moderate, heavy drinkers”). Why were these habits not taken into account on the binary backward logistic regression analysis? Is it because it was not significant on the long-term outcomes? If so, why not analysing the short-term and long-term outcomes separately in the backward multiple regression analysis?
- - The authors should mention in the abstract and the introduction that preclinical studies have shown controversial results on the combination of rt-PA and alcohol (Lemarchand et al., “Impact of alcohol consumption on the outcome of ischemic stroke and thrombolysis: role of the hepatic clearance of tissue-type plasminogen activator” Stroke 2015).
- - In the new paragraph concerning the effects of chronic alcohol intake and stroke outcomes, it would be necessary to include two studies reporting an aggravating effect of chronic alcohol consumption on the outcome of ischemic stroke (Ducroquet et al, Stroke 2013 and Drieu et al JCI Insight 2020).
Author Response
Dear Reviewer,
We greatly appreciate your efforts and time to evaluate our manuscript (jcm-2217924), entitled „Acute Alcohol Consumption Improves the Outcome of the Intravenous Thrombolysis Treatment in Acute Ischemic Stroke Patients- A Retrospective Study”. Each comment has been carefully considered again and the manuscript was revised accordingly. Point-by-point responses are indicated in bold and italics below.
Comments:
1/ As already mentioned in my first reviewing, I consider “acute alcohol consumption” in the title as misleading. “Positive blood alcohol content” seems to me a more accurate term. This is extremely relevant from a public health point of view.
Thank you for your comment. We changed “acute alcohol consumption” to “elevated blood alcohol content”.
2/ Still in the title, the term “improves” is, at least, an over interpretation and should be modified by “is associated with”.
Thank you for your comment. The new title is „Elevated Blood Alcohol Content Is Associated with Improved Clinical Outcomes of Intravenous Thrombolysis Treatment in Acute Ischemic Stroke Patients - A Retrospective Study.
3/ For me, the studied population is very different in terms of age and stroke severity from the most randomized, controlled trials on AIS and this fact needs to be mentioned, at least in the title and/or the abstract.
We accepted the reviewer’s suggestion and added the following sentence to the abstract: Patients were younger and had less severe stroke than in standard stroke studies (lines 34-35). As the inclusion of this sentence made the abstract longer, we had to shorten other parts in order to keep the 200 words limit. These minor modifications did not change the overall meaning of the abstract.
4/ There are mistakes in the OR values given in the abstract and the results sections concerning factors protecting against unfavourable short (OR:0.101) and long-term outcomes (0.182) compared to the values given in table 3 (0.091 and 0.187 respectively). Please look over.
Thank you very much for your thorough review, and noticing this typo, we corrected it.
5/ Alcohol consumption habits were significantly associated with the short-term outcome (supplementary table 1, p. 14) (by the way, it is not clear whether the significance (p=0.003) concerns “non-, mild drinkers” or “moderate, heavy drinkers”). Why were these habits not taken into account on the binary backward logistic regression analysis? Is it because it was not significant on the long-term outcomes? If so, why not analysing the short-term and long-term outcomes separately in the backward multiple regression analysis?
The significance (p=0.003) mentioned concerns “non-, mild drinkers” vs. “moderate, heavy drinkers”. In Supplementary Table 1, respective fields in the Table were merged and cell alignment was modified to centre alignment to ensure clarity. As it was mentioned in the legend of Table 3., alcohol consumption habits were taken into account in the binary backward multiple regression analysis, separately for the short-term and long-term outcomes.
“Last step of backward multiple regression analysis is provided. £Unfavorable short-term outcome is defined as a less than 4 points decrease of NIHSS and the NIHSS>0 by day 7 post-event. Backward multiple regression models included age, sex, acute alcohol consumption, hypertension, antidiabetic therapy, lipid lowering therapy, NIHSS at admission, CRP at admission, AST at admission, GGT at admission, MCV at admission (threshold>100 fL), admission systolic and diastolic blood pressure, alcohol consumption habits (non- or mild drinker vs. moderate or heavy drinker), presence of dysarthia at admission and 24 hours. §Unfavorable long-term outcome is defined as mRS 3-6 by day 90 post-event. Backward multiple regression models included age, sex, acute alcohol consumption, hypertension, diabetes mellitus, NIHSS on admission, admission serum glucose level, CRP at admission, WBC at admission, AST at admission, GGT at admission, MCV at admission (threshold>100 fL), admission systolic and diastolic blood pressure, alcohol consumption habits (non- or mild drinker vs. moderate or heavy drinker).”
6/ The authors should mention in the abstract and the introduction that preclinical studies have shown controversial results on the combination of rt-PA and alcohol (Lemarchand et al., “Impact of alcohol consumption on the outcome of ischemic stroke and thrombolysis: role of the hepatic clearance of tissue-type plasminogen activator” Stroke 2015).
Thank you for your comment. We added the following sentence to the introduction: Lemarchand et al. found that 6 weeks of alcohol consumption in a murine model influenced the hepatic clearance of tPA, leading to worsening of ischemic lesion and lack of beneficial effect of tPA induced thrombolysis (lines 53-55).
7/ In the new paragraph concerning the effects of chronic alcohol intake and stroke outcomes, it would be necessary to include two studies reporting an aggravating effect of chronic alcohol consumption on the outcome of ischemic stroke (Ducroquet et al, Stroke 2013 and Drieu et al JCI Insight 2020).
Thank you for your suggestion. We included the 2 studies, and added the following sentence to the discussion: Others also reported that chronic heavy alcohol drinking has an aggravating effect on the outcome of ischemic stroke (lines 313-314).
Thank you again for your thorough review of our work and the important comments raised. We believe that the changes resulted in an improvement of our manuscript. We hope that the revised manuscript will be found to be acceptable for publication in the ’Journal of Clinical Medicine’.
Yours sincerely,
László Oláh
corresponding author
olah@med.unideb.hu
